# Sarcopenic Obesity in Non-Alcoholic Fatty Liver Disease—The Union of Two Culprits

**DOI:** 10.3390/life11020119

**Published:** 2021-02-04

**Authors:** Saad Emhmed Ali, Mindie H. Nguyen

**Affiliations:** 1Department of Medicine, Division of Gastroenterology and Hepatology, Stanford University Medical Center, Palo Alto, CA 94304, USA; saadali@stanford.edu; 2Department of Epidemiology and Population Health, Division of Gastroenterology and Hepatology, Stanford University Medical Center, Palo Alto, CA 94304, USA

**Keywords:** sarcopenia, non-alcoholic fatty liver disease, obesity, insulin resistance, prevalence, metabolic diseases

## Abstract

Non-alcoholic fatty liver disease (NAFLD) continues to rise and has become the most common cause of chronic liver disease among all ages and ethnicities. Metabolic disorders, such as obesity and insulin resistance, are closely associated with sarcopenia and NAFLD. Sarcopenic obesity is a clinical disorder characterized by the simultaneous loss of skeletal muscle and gain of adipose tissue. It is associated with worse outcomes in individuals with NAFLD. It is projected that NAFLD and sarcopenia will rise as the prevalence of obesity continues to increase at an unparallel rate. Recently, sarcopenia and sarcopenic obesity have gained considerable interest, but we still lack a well-defined definition and a management approach. Therefore, it is imperative to continue shining the light on this topic and better understand the underlying mechanism as well as treatment options. In this review article, we aimed to address the pathophysiology, impact, and outcomes of sarcopenic obesity on NAFLD.

## 1. Introduction

Non-alcoholic fatty liver disease (NAFLD) is the most common cause of chronic liver disease with a reported global prevalence of 25%, with regional variation of about 30% for the United States, South America, the Middle East, 24% for Europe, and 14% for Africa [1,2,3]. NAFLD includes a clinicopathologic continuum, from an excess of hepatic steatosis to non-alcoholic steatohepatitis (NASH), progressive fibrosis, cirrhosis, liver cancer, end-stage liver disease, requiring liver transplantation, and death [4,5,6]. NAFLD is commonly noticed in obese but also in nonobese individuals with metabolic derangement such as visceral obesity and insulin resistance, both of which closely implicated in sarcopenia “muscle wasting” [7,8,9].

Sarcopenia is identified as a progressive and diffuse decline in skeletal muscle mass, function, and strength [10]. Insulin resistance, vitamin D deficiency, and chronic systemic inflammation are common pathogenetic pathways for both sarcopenia and NAFLD [11,12]. Sarcopenic obesity is a term used to describe the coexistence of sarcopenia and obesity [13,14]. Sarcopenia and sarcopenic obesity were first observed to be entangled in NAFLD in the Korean Sarcopenic Obesity study by Hong et al., who found that people with higher body mass index [BMI] and fat mass had an increased prevalence of NAFLD [15]. Since then, this observation has gained significant interest, and sarcopenic obesity has been found to be associated with a higher risk of metabolic syndrome, cardiovascular disease, liver disease, frailty, mobility disability, healthcare expenses, and mortality in the ambulatory population [16,17]. Among inpatients with end-stage liver disease waiting for liver transplantation, the prevalence of sarcopenic obesity can be up to 35% [18]. In this current review, we aim to present an up-to-date overview of the definitions, diagnostic methods, pathophysiology, impacts, and management approaches for sarcopenic obesity for individuals with NAFLD.

## 2. Definitions and Diagnostic Modalities

Sarcopenia was first termed in 1989 by Rosenberg et al. as a muscle loss in the aging population [19]. Although several definitions coexist, sarcopenia’s well-adopted description is an age-related process characterized by progressive loss of skeletal muscle mass and function [10]. Several assessment tools can be utilized to assess muscle mass and function (Table 1). To determine the muscle mass, anthropometric measurements, such as the thickness of the skin fold, mid-upper arm muscle circumference (MAMC), and calf circumference have been tried in the past and found to be inaccurate; therefore, these methods are not recommended [20]. Imaging such as computed tomography (CT) at the level of the third lumbar vertebra to assess skeletal muscle index (SMI) (cut-off value of <50 cm^2^/m^2^ in men and <39 cm^2^/m^2^ in women) and magnetic resonance imaging (MRI) have been used effectively to differentiate between muscle, fat and other soft tissues. However, due to risk of radiation and/or cost, these tests have not been widely used in routine clinical practice [20,21,22] (Figure 1). Instead, dual-energy X-ray absorptiometry (DEXA) is generally recognized as the gold assessment tool to assess the muscle mass in most cases, due to its low cost, low radiation risk of radiation, and reasonable accuracy [23]. Bioimpedance analysis (BIA) can be also used as an affordable alternative method to DEXA [23]. Lastly and importantly, it should be noted that the accuracy of both BIA and DEXA in measuring muscle mass requires patients to be well hydrated prior to undergoing these tests.

In regard to muscle function assessment, there are several proposed methods, but the most common practice is to measure hand-grip strength [24]. The short physical performance battery (SPPB), gait speed, timed get-up-and-go test are other tests to measure the physical performance of muscle [20,25].

The definition of obesity should also be revisited for a number of reasons. While obesity is generally known as a condition of excessive fat deposition and routinely evaluated in clinical practice using body mass index (BMI), waist circumference, and DEXA, there are limitations to these techniques. First, BMI does not differentiate between fat or lean body mass. Similarly, BMI does not differentiate between “dry” euvolemic weight and “wet” hypervolemic individuals such as those with end-stage liver disease (ESLD). In addition, the diagnostic cut-off for obesity adults is race/ethnic dependent [26]. Obesity and overweight are defined as BMI > 30 and 25 kg/m^2^ in non-Asians and >27.5 and 23.0 kg/m^2^ in Asians, respectively, as complications of obesity occur at a lower BMI in Asians [26]. Additional education efforts may be needed to raise the awareness of these racial/ethnic variation since a recent U.S. population-based study reported the highest rate of discordance between self-perception of overweight and actual weight among Asians with NAFLD [27].

Finally, sarcopenic obesity describes a combination of skeletal muscle loss and fat gain, initially observed by Baumgartner et al.; this condition is commonly found in an elderly population [28]. With aging, muscle mass decreases and adipose tissue increases; by the third decade of life, muscle mass decreases by about 1% per year [28,29]. Sarcopenic obesity has been shown to associate with higher morbidity and mortality than either obesity or sarcopenia alone. Comorbidities that have been described to associate with sarcopenic obesity include metabolic syndrome, insulin resistance, hypertension, and cardiovascular disease [30,31].

## 3. Epidemiology, Prevalence, and Risk Factors

As the body composition of fat and muscle change with aging, there are expected physiologic changes with age. Body fat increases with age and peaks in the seventh decade of life and then declines [32,33]. On the other hand, muscle mass decreases with age starting in the fourth decade [34]. In the cases of pathologic muscle loss and pathologic fat accumulation, there are no well-defined cut-off values, which consequently give rise to wide variation in the reported prevalence of this condition [25,28,31,35,36,37,38,39,40]. The prevalence values range from 2% to 25% in studies used bioelectrical impedance analysis (BIA) and anthropometric measurements [41,42,43]. In studies using DEXA, the reported prevalence was higher with values ranging from 2.7% to 84% [25,28,38,40,44,45,46,47,48]. In a recent analysis of 270 individuals aged 65 years or older who were enrolled in the Frailty in Brazilian Older People Study, the prevalence of sarcopenic obesity was 29.3% with sarcopenic obesity diagnosed via DEXA and defined as a body fat percentage ≥38% or ≥27% and appendicular skeletal muscle mass index (ASMMI) of <5.45 or <7.26 kg/m2 in men or women, respectively [49].

The risk factors for sarcopenic obesity include age-related decrease in the resting metabolic rate due to low physical activity, decreased mitochondrial volume and its oxidative capacity [50,51,52,53]. Sex-related hormonal changes in estrogen; progesterone and testosterone also affect the muscle and fat composition. For example, in post-menopausal women, the body fat, in particular the visceral fat, increases and the muscle mass decreases as a result of low estrogen levels [54]. In men, testosterone stimulates muscle regeneration, and its deficiency which decreases by about 1% each year can promote low muscle mass and high fat mass [55,56]. Other risk factors are sedentarism, ethnicity with higher sarcopenic obesity prevalence in Hispanic and non-Hispanic whites compared to non-Hispanic blacks [48,57]. Finally, sarcopenic obesity was also found to be highly prevalent in patients with advanced chronic kidney disease (CKD) such as CKD stage 4 [58].

## 4. The Pathogenesis of Sarcopenic Obesity and NAFLD

Several pathophysiological mechanisms, such as insulin resistance, high pro-inflammatory cytokines, decreased exercise, and hormonal changes have been described as common pathways for both NAFLD and sarcopenic obesity [59]. The increased expression of pro-inflammatory genes can play a significant role in the recruitment of immune cells into the adipose tissue during weight gain [60]. The secretion of inflammatory mediators or the recruitment of macrophages is strongly linked to the remodeling of adipose tissue and to the pathogenesis of co-morbidities during obesity [60]. The muscle-liver-adipose tissue axis has been suggested as a leading cause of NAFLD and sarcopenic obesity, as shown in Figure 2 [61,62,63], though the exact interplay between skeletal muscle, liver, and adipose tissue is still not yet fully understood. Adipose tissue has been recognized as an endocrine organ that secretes hormones called adiponectin and leptin, both of which play a vital role in regulating insulin sensitivity, glucose, and lipid metabolism [61,64]. First, adiponectin enhances insulin sensitivity and has antifibrotic, anti-inflammatory, and antisteatotic effects on the liver [64,65]. Therefore, the reduction in the adiponectin level leads to insulin resistance and glucose intolerance and may lead to hepatic injury and fatty liver [64,65]. Second, leptin stimulates hepatic stellate cells, which in turn enhances the process of fibrogenesis and inflammation in the liver [61,66]. Once the hepatic stellate cell is activated, it can also produce more leptin and further strengthen the fibrogenesis cascade, eventually leading to more liver fibrosis and cirrhosis [61,66]. Furthermore, leptin promotes the progression of liver fibrosis by stimulating the synthesis of transforming growth factor beta (TGFß) in the Kupffer cells [61].

Skeletal muscles also act as an endocrine organ via the autocrine pathways and secrete important myokines [67]. One of these myokines is called myostatin, a TGFß superfamily member, an essential regulator of the skeletal muscles, adipose tissue, and liver metabolisms [67]. Myostatin has a lipogenesis effect that increases adipose tissue mass and lowers adiponectin secretion. In addition, myostatin has a proteostasis effect that decreases the skeletal muscle mass [61,67]. In animal model studies, blocking the function of myostatin has been shown to lower insulin resistance, enhance skeletal muscle mass, and protect against the development of fatty liver [68,69]. The trigger for this muscle–liver–adipose tissue interplay was speculated to begin in the skeletal muscle with downstream influence on both adipose tissue and the liver [61]. The other important myokine is interleukin 6 (IL-6), which plays a potential role in regulating fatty acid oxidation in the liver and insulin resistance [61]. Secreted factors by adipose tissue and skeletal muscle strongly regulate adipocyte browning [70]. For example, physical activity regulates the differentiation of these “anti-obesity” adipocytes by releasing irisin or IL-6 [70]. It was also shown that IL-6 secreted by brown or beige adipocytes positively affects glucose homeostasis [71,72]. During sarcopenia, the positive effects of physical exercise and thermogenesis can be attenuated, which can significantly contribute to NAFLD progression.

## 5. Sarcopenic Obesity Impact in Non-Alcoholic Fatty Liver Disease

Sarcopenia and obesity are associated with a higher risk of cardiovascular and metabolic diseases and higher healthcare-related expenditure and mortality in the elderly population [16,17,73]. Obesity is associated with higher level of liver fibrosis and worse prognosis in patients with NASH [74], and sarcopenia was found to be one of the culprits contributing to the progression of NAFLD-associated fibrosis [75]. In a study of 225 patients with histologically proven NASH by Petta et al., the presence of sarcopenia was associated with more severe fibrosis [63]. Another study by Lee et al. revealed a strong association between sarcopenia and advanced hepatic fibrosis in NAFLD patients that was independent of the other co-factors, such as insulin resistance and obesity [9]. In another study of 309 patients with biopsy-proven NAFLD by Koo et al., sarcopenic patients had a higher likelihood of developing NASH than non-sarcopenic patients (odds ratio 2.28; 95% confidence interval 1.21–4.30) [8]. The risk of all-cause mortality was also shown to be increased by 24% in sarcopenic obese individuals compared with those without sarcopenic obesity in a large meta-analysis by Tian et al. [17]. The negative impact of sarcopenic obesity was not only limited to people with NAFLD but also in patients with cancers who were found to have higher risk of complications related to treatments, poor physical performance and shorter survival if sarcopenic obesity was present [76].

Obesity and NAFLD are both major risk factors for the development of hepatocellular carcinoma (HCC) [77]. A meta-analysis investigating the association between obesity and HCC revealed that obese individuals had an 89% higher risk of HCC than normal-weight individuals [78]. Moreover, obese individuals have higher cancer-related mortality than their normal-weight counterparts [79]. Another systemic review and meta-analysis of nine observational studies involving more than 1.5 million individuals revealed that obese patients had a two-fold rise in HCC-related mortality, in particular men and patients from Western countries [79]. Furthermore, Kobayashi and colleagues recently examined the effect of sarcopenic obesity in patients undergoing surgical resection for HCC and found that sarcopenic obese patients had lower 3-year recurrence-free survival than non-sarcopenic obese patients (19.3% vs. 37.8%, respectively; *p* = 0.003) as well as lower over-all survival (45.6% vs. 61.0%, respectively; *p* = 0.002) [80].

## 6. Conclusions

Sarcopenic obesity is a co-occurrence of two significant comorbidities with a higher risk of morbidity and mortality than either obesity or sarcopenia alone. It is a relatively new concept that is increasingly recognized to occur in a significant proportion of the population and associate with poor outcomes, especially in the elderly population and those with NAFLD. There is a vicious cycle in which sarcopenic obesity and NAFLD contribute to each other through the complex interplay of the muscle–adipose–tissue–liver axis. A clear consensus on the definition of sarcopenic obesity is needed to encourage more timely diagnosis, to better inform clinical management, and to facilitate future research.

## Figures and Tables

**Figure 1 life-11-00119-f001:**
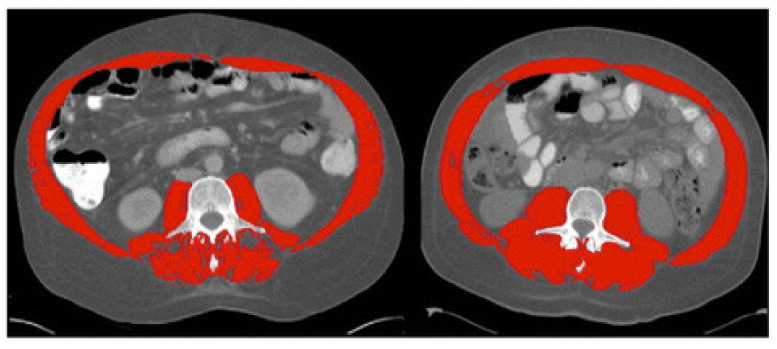
Computed tomography (CT) images at the third lumbar vertebra comparing two patients with cirrhosis and same body mass index but one has sarcopenia and other is not. Left image shows sarcopenia with L3mi of 49.82 cm^2^/m^2^. Right image shows non-sarcopenia with L3mi of 70.8 cm^2^/m^2^. These abdominal CT images were utilized by Montano-Loza et al. in 2012 and cutoff-values offered by Baumgartner et al. in 1998. Red color denotes abdominal wall muscle, psoas and paraspinal muscles. (Used with permission from Elsevier.).

**Figure 2 life-11-00119-f002:**
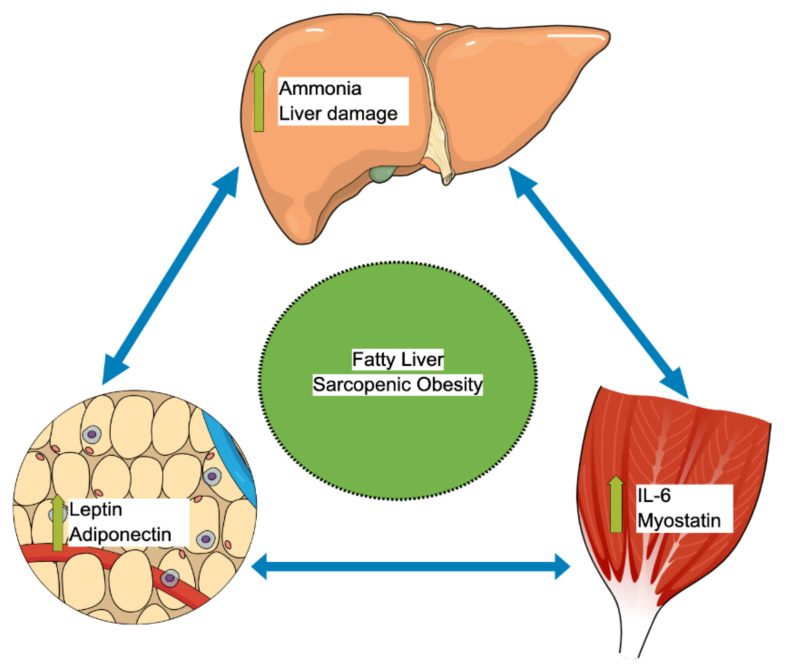
Muscle-liver-adipose tissue axis autocrine interplay that leads to sarcopenic obesity and fatty liver. IL-6: Interleukin-6.

**Table 1 life-11-00119-t001:** Commonly used tests to assess sarcopenia (muscle mass and function).

Muscle Mass	Muscle Function
L-3 Computed tomography (CT) (skeletal muscle index)	Handgrip strength (HGS)
Magnetic resonance imaging (MRI)	Short physical performance battery (SPPB)
Dual-energy X-ray absorptiometry (DEXA)	Gait speed
Bioimpedance analysis (BIA)	Timed get-up-and-go test

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
