# Peer review of "Sarcopenic Obesity in Non-Alcoholic Fatty Liver Disease—The Union of Two Culprits"

_life, 2021, doi:10.3390/life11020119_

Round 1
Reviewer 1 Report
The manuscript is a comprehensive review which focuses on a clinically important issue. The relevant findings are discussed in a logical order; however, some critical elements are missing.
Major comments:
- Increased expression of pro-inflammatory genes can play a significant role in the recruitment of immune cells into the adipose tissue during weight gain. The secretion of inflammatory mediators or the recruitment of macrophages is strongly linked to the remodeling of adipose tissue and to the pathogenesis of co-morbidities during obesity (e.g. PMID: 15663921).
- Secreted factors by adipose tissue and skeletal muscle strongly regulate adipocyte browning, e.g. physical activity regulates the differentiation of these “anti-obesity” adipocytes through the release of myokines, e.g. irisin or IL-6 (PMID: 32236810). It was also shown that IL-6 secreted by brown or beige adipocytes has a beneficial effect on glucose homeostasis (PMID: 23221344, 30794803). During sarcopenia, the positive effects of physical exercise and thermogenesis can be attenuated, which can significantly contribute to the progression of NAFLD.
Minor comments:
- The abbreviation ESLD should be defined (page 2).
- The muscle-liver-adipose tissue axis involves endocrine instead of autocrine pathways (page 3).
Author Response
Response to Reviewer 1 Comments:
The manuscript is a comprehensive review which focuses on a clinically important issue. The relevant findings are discussed in a logical order; however, some critical elements are missing.
Major comments:
- Increased expression of pro-inflammatory genes can play a significant role in the recruitment of immune cells into the adipose tissue during weight gain. The secretion of inflammatory mediators or the recruitment of macrophages is strongly linked to the remodeling of adipose tissue and to the pathogenesis of co-morbidities during obesity (e.g. PMID: 15663921).
Response: Thank you for your comments. We have added this to the manuscript (page 3, lines 26-30).
- Secreted factors by adipose tissue and skeletal muscle strongly regulate adipocyte browning, e.g. physical activity regulates the differentiation of these “anti-obesity” adipocytes through the release of myokines, e.g. irisin or IL-6 (PMID: 32236810). It was also shown that IL-6 secreted by brown or beige adipocytes has a beneficial effect on glucose homeostasis (PMID: 23221344, 30794803). During sarcopenia, the positive effects of physical exercise and thermogenesis can be attenuated, which can significantly contribute to the progression of NAFLD.
Response: Thank you for your comments. We have added this to the manuscript (page 4, lines 2-7).
Minor comments:
- The abbreviation ESLD should be defined (page 2).
Response: Thank you for your comments. We have clarified the abbreviation as suggested (page 2, line 34).
- The muscle-liver-adipose tissue axis involves endocrine instead of autocrine pathways (page 3)
Response: We have made the correction (page 3, line 30,31). Thank you.
Reviewer 2 Report
This manuscript was well organized, comprehensively described the pathophysiology, impact, and outcomes of sarcopenic obesity on NAFLD. NAFLD is a health problem of increasing importance in recent years. In this respect, the research on the relationship with sarcopenia and/or obesity is considered to be a very important field. In addition, this paper properly explained about the association between sarcopenia without obesity and NAFLD and/or obesity and NAFLD and the difference between sacopenic obesity.
However, it is desirable to present research results on these disease states and aspects of liver cancer carcinogenesis and cancer prognosis.
Additionally, it is necessary to unify the reference form.
Author Response
Response to Reviewer 2 Comments:
Comments and Suggestions for Authors
This manuscript was well organized, comprehensively described the pathophysiology, impact, and outcomes of sarcopenic obesity on NAFLD. NAFLD is a health problem of increasing importance in recent years. In this respect, the research on the relationship with sarcopenia and/or obesity is considered to be a very important field. In addition, this paper properly explained about the association between sarcopenia without obesity and NAFLD and/or obesity and NAFLD and the difference between sarcopenic obesity.
However, it is desirable to present research results on these disease states and aspects of liver cancer carcinogenesis and cancer prognosis. Additionally, it is necessary to unify the reference form.
Response: Thank you for your comments. We have added the following revision as suggested (page 4, lines 26-37).
Obesity and NAFLD are both major risk factors for the development of hepatocellular carcinoma (HCC) [77]. A meta-analysis investigating the association between obesity and HCC revealed that obese individuals had an 89% higher risk of HCC than normal-weight individuals [78]. Moreover, obese individuals have higher cancer-related mortality than their normal-weight counterparts [79]. Another systemic review and meta-analysis of 9 observational studies involving more than 1.5 million individuals revealed that obese patients had a two-fold rise of HCC-related mortality, in particular men and patients from Western countries [79]. Furthermore, Kobayashi and colleagues recently examined the effect of sarcopenic obesity in patients undergoing surgical resection for HCC and found that sarcopenic obese patients had lower 3-year recurrence-free survival than non-sarcopenic obese patients (19.3% vs. 37.8%, respectively; P=0.003) as well as lower over-all survival (45.6%vs. 61.0%, respectively; P=0.002) [80].
Reviewer 3 Report
In recent years, nonalcoholic fatty liver disease ( NAFLD) have replaced chronic hepatitis C alongside alcoholic liver damage as the most common chronic liver disease in the western world. NAFLD encompass a broad spectrum of liver diseases, ranking from nonalcoholic fatty liver (NAFL), nonalcoholic steatohepatitis ( NASH), secondary fatty liver to fatty cirrhosis. Several factors are associated with a higher risk of developing an NAFLD, especially, NASH. These include insulin resistance, central obesity, genetic and environmental factors, and changes in intestinal flora. The rapid rise in NAFLD is closely linked to the large increase in metabolic mulitisystemic diseases. NASH has a multifactorial genesis within the scope of which, in addition to genetic and lifestyle factors (overaeting anf malnutrition), mitochondrial dsyfunctions, endotoxins and proinflammatory cytokines contribute to chronic inflammation. Sacrcopenia is charactreised by a progressive and generalised decrease in muscle mass and strength, combined with a progressive loss of function. It occurs in approximately 10 - 15 % of all adults over the age of 65 as part of the physiological ageing process, during which hormonal changes, metabolic factors and reduced activity are of particular significance. This process begins its gradual after the 50. Sarcopenia is often closely associated with chronic diseases, in particular cirrhosis, as well as the fate and prognosis of patients. A joint occurrence of an NAFLD and sarcopenia is growing in significance with respect to morbidity and mortality rates, thus rendering investigations into the question of pathogenesis, the opportunities prevented by early diagnosis and effective therapy and / or prevention highly relevant and novel. Insulin resistance, the effect of proinflammatory cytokines, especially IL-6, hormonal changes and physiological inactivity rate are at the centre of pathophysiological mechanisms " sarcopenic obesity". With both diseases, discussion also considers a lack of micronutrients, such as zinc and vitamin D, particularly to chronic inflammation ( 1, 2, 3).
Following a brief and practical introduction, giving the definition of the new disease entity, the authors will provide an effective overview of the epidemiology, prevalence and risk factors of the disease. The following sections contains a presentation of the various interacting pathogenetic mechanisms. Finally, the authors demonstrate that this combination of sarcopenia and obesity with an NAFLD is not only closely associated with an increased risk of cardiovascular diseases, but also in particular an increased degree of liver fibrosis and poorer prognosis of NASH.
The paper provides an effective overview of key aspects of the co-existence of sarcopenia and an NAFLD. The statements are factual and well substantiated by a suitable collection of literature ( 72 citations). The paper is written in correct and legible English. The figure and the table provide an effective complement to the text explanations.
I recommend the acceptance of the paper in the present form.
Literature
- Nishikawa H. et al. Serum zinc concentrations and sarcopenia: aclose linkage in chronic liver diseases. J Clin Med 2019; 8: doi: 10339/jcm8030336.
- Schiawo L. et al. Nutritional status in patients with obesity and cirrhosis. Worl J Gastroenterol 2018; 24: 3330 - 3346.
- Grüngreiff K; Anand A. Branched chain amino acids and zinc in th nutrition of liver cirrhosis. J Clin Exp Hepatol 2018; 8: 480 - 483.
Author Response
Response to Reviewer 3 Comments:
I recommend the acceptance of the paper in the present form.
Response: Thank you so much for your kind comments.
Round 2
Reviewer 1 Report
The Manuscript was substantially improved, based on the comments of the reviewers. Therefore, I suggest the current version of the Manuscript for publication in this Journal.